

# A development of cloud top height retrieval using thermal infrared spectra observed with GOSAT and comparison with CALIPSO data

Y. Someya[1], R. Imasu[1], N. Saitoh[2], Y. Ota[3], and K. Shiomi[4]

[1]Atmosphere and Ocean Research Institute, University of Tokyo, Chiba, Japan
[2]Center of Environmental Remote Sensing, Chiba University, Chiba, Japan
[3]Meteorological Research Institute, Tsukuba, Japan
[4]Japan Aerospace eXploration Agency, Tsukuba, Japan

Received: 28 November 2015 – Accepted: 11 December 2015 – Published: 14 January 2016

Correspondence to: Y. Someya (y_someya@aori.u-tokyo.ac.jp)

Published by Copernicus Publications on behalf of the European Geosciences Union.

**AMTD**

doi:10.5194/amt-2015-371

A development of cloud top height retrieval

Y. Someya et al.



## Abstract

An algorithm based on $CO_2$ slicing, which has been used for cirrus cloud detection using thermal infrared data, was developed for high-resolution radiance spectra from satellites. The channels were reconstructed based on sensitivity height information of
the original spectral channels to reduce the effects of measurement errors. The selections of the reconstructed channel pairs were optimized for several atmospheric profile patterns using simultaneous studies assuming cloudy sky. That algorithm was applied to data by the Greenhouse gases Observing SATellite (GOSAT). The results were compared with those obtained from space-borne lidar instrument onboard Cloud–Aerosol
Lidar and Infrared Pathfinder Satellite Observations (CALIPSO). Monthly mean cloud amounts from the slicing generally agreed with those from CALIPSO observations despite some differences caused by surface temperature biases, optically very thin cirrus, multilayer structures of clouds, extremely low cloud tops, and specific atmospheric conditions. Comparison of coincident data showed good agreement except some cases
and revealed that the improved slicing method is more accurate than the traditional slicing method. Results also imply that improved slicing can detect low-level clouds with cloud top heights as low as approximately 1.5 km.

## 1  Introduction

Global warming is well known to have been caused by increasing greenhouse gas
(GHG) emissions since the Industrial Revolution in the eighteenth century. The concentrations of $CO_2$ and $CH_4$, as the main GHGs occupying about 80 % of the greenhouse effect, are 396.0 ppm and 1824 ppb, accounting for 142 and 253 % of the level before the Industrial Revolution. They are still increasing 2.07 ppm and 3.8 ppb per year, respectively, in this decade (WMO, 2014). Although GHGs have been measured
mainly using ground-based observations, the sites are regionally limited. The Greenhouse gases Observing SATellite (GOSAT) was launched in 2009 to monitor these

Discussion Paper | Discussion Paper | Discussion Paper | Discussion Paper |

# AMTD

doi:10.5194/amt-2015-371

## A development of cloud top height retrieval

Y. Someya et al.

GHGs, supporting global observations. The instrument on-board GOSAT is called the Thermal And Near-infrared Sensor for carbon Observation (TANSO), which consists of a Fourier Transform Spectrometer (FTS) and a Cloud and Aerosol Imager (CAI). FTS, the main sensor for gas retrieval, has three bands in the Short Wave InfraRed (SWIR) region and one band in the Thermal InfraRed (TIR) region. In addition, CAI has four bands in the wavelength range from ultraviolet to near-infrared observes clouds and aerosols, which prevent gas retrieval and FTS data being judged in terms of whether the scene is clear, based on CAI observations. The cloud detection algorithm for analyzing observation data from CAI is called the Cloud and Aerosol Unbiased Decision Intellectual Algorithm (CLOUDIA; Ishida and Nakajima, 2009). This algorithm calculates the confidence clear probability for each pixel with thresholds based on sensitivity tests. The GHGs are retrieved from FTS data only if all the pixels in CAI pixels corresponding to the instantaneous field of view (IFOV) of FTS are clear. CAI has horizontally high resolution and it enables to detect partial cloud within the IFOV. However, this algorithm presents some weaknesses: it is difficult to distinguish clouds and high reflectivity surfaces; also, optically thin clouds are detected only to a slight degree.

The columnar averaged concentrations of $CO_2$ and $CH_4$ retrieved from SWIR data are mainly validated with those obtained from the ground-based observation network called the Total Carbon Column Observing Network (TCCON; Wunch et al., 2011). Reportedly, SWIR Level 2 V01.xx products had biases of $-8.85$ ppm for $CO_2$ and $-20.4$ ppb for $CH_4$ compared with TCCON observations (Morino et al., 2011). These biases are probably attributable to the existence of optically thin clouds or aerosols. Uchino et al. (2012) demonstrated that these biases can be reduced if they are retrieved with consideration of accurate cloud and aerosol properties observed from ground-based lidar resources. In the algorithm of SWIR L2 V02.xx, the aerosol optical thickness estimated from SWIR band data were considered in the retrievals. The biases were reduced to $-1.48$ ppm for $CO_2$ and $-5.9$ ppb for $CH_4$ (Yoshida et al., 2013). However, the biases were not removed completely. To elucidate the effects of clouds and aerosols in the gas retrievals, their altitude information must be known. Moreover,

**AMTD**

doi:10.5194/amt-2015-371

**A development of cloud top height retrieval**

Y. Someya et al.

more accurate cloud information such as altitude reduces bias. The cloud altitudes must be estimated from FTS data because CAI has no sensitivity to them.

Vertical distributions of $CO_2$ and $CH_4$ at the upper troposphere are estimated from FTS Band 4 (TIR band) data (Saitoh et al., 2009). Actually, TIR data are obtained for the entire day, but CAI observes only in the daytime. Therefore, clouds must be detected using TIR data in the nighttime. Current cloud retrieval techniques used with TIR data from GOSAT discriminate clouds from the surface using brightness temperature contrast at the atmospheric window region near 10 µm (Imasu et al., 2010). However, this technique, calling TIR threshold technique here, detects optically thin clouds or partly existing clouds in the IFOV only to a slight degree. Consequently, more accurate cloud detection methods must incorporate GOSAT retrieval to detect optically thin clouds such as cirrus. Moreover, such methods can improve the gas retrieval data quality. The $CO_2$ slicing method, which was developed as a detection technique for optically thin clouds (Chahine et al., 1974; Smith and Platt, 1978; Menzel et al., 1983), can overcome these limitations. This method has been used to derive high-level cloud climatology with a thermal infrared sounder such as Visible Infrared Spin-Scan Radiometer Atmospheric Sounder (VAS), High Resolution Infrared Radiometer Sounder (HIRS), and MODerate resolution Imaging Spectroradiometer (MODIS) (Smith and Platt, 1978; Menzel et al., 1983, 1992; Wylie and Menzel, 1989, 1999; Wylie et al., 1994, 2005; Chang et al., 2010). However, this method is effective only for high-level clouds because of the small brightness temperature contrast between clouds and surfaces and because most absorption bands have sensitivity at high levels. Because GOSAT obtains high-resolution spectra, several channels in the $CO_2$ absorption region have sensitivities at the low or middle level of the atmosphere. Therefore, middle or low-level cloud detection is possible from this technique using spectral data in this region. Holtz et al. (2006) presented an improvement of the slicing method called "Sorting–Slicing" for spectral data from the Scanning High-Resolution Interferometer Sounder (S-HIS). Nevertheless, few reports describe modification of the slicing method for application to spectral data from satellites.

**AMTD**

doi:10.5194/amt-2015-371

**A development of cloud top height retrieval**

Y. Someya et al.

Discussion Paper | Discussion Paper | Discussion Paper | Discussion Paper

This paper presents improvement of the $CO_2$ slicing method. Section 2 describes the satellite products, atmospheric parameter datasets, and radiative transfer codes for radiative transfer simulations. Section 3 presents a description of the cloud retrieval algorithm, which is based on standard $CO_2$ slicing, and improvements with channel reconstruction and optimization of channel pairs to reduce detection errors based on simulation studies assuming several atmospheric conditions. This improved algorithm was applied to TIR spectra from GOSAT in Sect. 4. Derived cloud amounts are compared with those from Cloud–Aerosol Lidar and Infrared Pathfinder Satellite Observations (CALIPSO) observations statistically in Sect. 4.1. Cloud top heights from coincident observation data of GOSAT and CALIPSO are compared in Sect. 4.2.

## 2 Datasets and radiative transfer models

GOSAT, in an approximately 666 km height sun-synchronous polar orbit with a revisit cycle of 3 days and equator-crossing time of 13:00 LT, covers almost the entire Earth between about 85° N and 85° S. The TANSO-FTS instrument has four bands in the range of 0.758–0.775 μm (Band 1), 1.56–1.72 μm (Band 2), and 1.92–2.08 μm (Band 3) in the SWIR region and also 5.5–14.3 μm (Band 4) in the TIR region with spectral resolution of about $0.2\,cm^{-1}$. Its size of IFOV is 15.8 mrad, which corresponds to diameter of approximately 10.5 km at the earth's surface. TANSO-FTS has many observation patterns. The maximum pointing angle is ±35° in a cross-track direction and ±20° in an along-track direction (Kuze et al., 2009). The spectral data of FTS band 4 Level 1B V150.151 products observed in 2010 (Kuze et al., 2012) provided from NIES were used for this study. The radiometric accuracy of this product is near 0.5 K in the range of $700–755\,cm^{-1}$ (Kataoka et al., 2014). Observational patterns in 2010 included five-point cross track scan mode until July and three-point cross-track scan mode from August. TANSO-CAI is an imager to discriminating clouds and aerosol within the IFOV of TANSO-FTS. It has four bands from the ultraviolet to near-infrared region, respectively

Discussion Paper | Discussion Paper | Discussion Paper | Discussion Paper |

**AMTD**

doi:10.5194/amt-2015-371

**A development of cloud top height retrieval**

Y. Someya et al.

centered at 0.380 μm (Band 1), 0.674 μm (Band 2), 0.870 μm (Band 3), and 1.60 μm (Band 4) with spatial resolution of 0.5 km (Band 1–3), and 1.5 km (Band 4) for pixels.

Although FTS and CAI are passive sensors using thermal or solar radiation, the most accurate measurement of clouds and aerosols are using an active sensor, light detection and ranging (lidar), which emits a visible or near-infrared laser beam and receives their back-scattered components. Its detection accuracy is higher than those of passive sensors. Lidar observation can estimate the vertical distribution of the back scattering coefficient, extinction coefficient, and depolarization ratio of cloud or aerosol layers accurately, even for optically very thin targets. Consequently, the analysis results of GOSAT data can be validated using data from the space-borne lidar, Cloud–Aerosol Lidar with Orthogonal Polarization (CALIOP), on CALIPSO, which was regarded as providing more reliable values. CALIPSO is also a sun-synchronous polar orbit satellite with a revisit cycle of 16 days. Its equator-crossing time is 13:30 LT. It covers between approximately 82° N to 82° S. CALIOP is a lidar system using laser wavelengths of 532 and 1064 nm. Vertical resolutions of sampling are 30 m below 8.2 km and 60 m between 8.2 and 20.2 km. The lidar footprint is a circle with about 90 m diameter at the surface. The spatial interval of footprints is 333 m along a track. The CALIOP Level 2–5 km Cloud/Aerosol Layer V3.01 products were used for this study. These products include information related to cloud/aerosol layer such as the number of layers up to 10, geometrical cloud top and bottom height, and optical thickness of the layer with vertical resolution of 5 km.

Sea surface temperature (SST), pressure, temperature, and humidity from the surface to the 10 hPa pressure level are estimated using linear temporal and spatial interpolation of Global Spectral Model (GSM) – Grid Point Value (GPV) data provided by the Japan Meteorological Agency. These meteorological data were used as inputs to radiative transfer calculations for each observation. The GSM-GPV data are provided for four times per day with spatial resolution of 21 layers vertically and 0.5° × 0.5° horizontally. Surface emissivity is referred from the Advanced Spaceborne Thermal Emission

**AMTD**

doi:10.5194/amt-2015-371

**A development of cloud top height retrieval**

Y. Someya et al.

and Reflection Radiometer (ASTER) Spectral Library (Baldridge et al., 2009) based on the land cover type from International Geosphere–Biosphere Programme (IGBP).

The Line-By-Line Radiative Transfer Model (LBLRTM; Clough et al., 2005) provided by Atmospheric Environmental Research Inc. (AER) was used for radiative transfer calculations considering gas absorption based on the HIgh-resolution TRANsmission molecular absorption database (HITRAN) 2004 (Rothman et al., 2005). The Polarization System for Transfer of Atmospheric Radiation ver. 3 (Pstar3) (Ota et al., 2010) was used for theoretical radiative transfer calculations including multi-scattering by cloud or aerosol particles in simulation studies.

## 3 Methodology

### 3.1 $CO_2$ slicing method

TANSO-FTS (Band 4) has a spectral channel in the $CO_2$ absorption band near 15 μm. The $CO_2$ slicing method uses the difference of the absorption strength between a pair of channels in this region. The concept of the $CO_2$ slicing method can be formulated as

$$\frac{R_{\lambda_1} - R_{\lambda_1}^{\text{clr}}}{R_{\lambda_2} - R_{\lambda_2}^{\text{clr}}} = \frac{\alpha_1 \epsilon_{\lambda_1} \int_{p_s}^{p_c} t_{\lambda_1}(p)\,\mathrm{d}B_{\lambda_1}}{\alpha_2 \epsilon_{\lambda_2} \int_{p_s}^{p_c} t_{\lambda_2}(p)\,\mathrm{d}B_{\lambda_2}}, \tag{1}$$

where $R$ stands for the observed radiance, $R^{\text{clr}}$ denotes the calculated clear sky radiance, $\alpha$ signifies a cloud fraction in the IFOV, $\epsilon$ represents the cloud emissivity, $p_s$ and $p_c$ are pressure at the surface and the clouds top respectively, $t$ denotes the transmittance, $B$ is the Planck function, and subscript $\lambda$ denotes the spectral channel wavelength. If the two spectral channels $\lambda_1$ and $\lambda_2$ are sufficiently close, it can be assumed that the fractions and the emissivity are equal ($\alpha_1 \epsilon_1 \cong \alpha_2 \epsilon_2$). The value $\alpha \epsilon$, called the Effective Cloud Amount (ECA), corresponds to the coverage if clouds in the FOV are

**AMTD**

doi:10.5194/amt-2015-371

**A development of cloud top height retrieval**

Y. Someya et al.

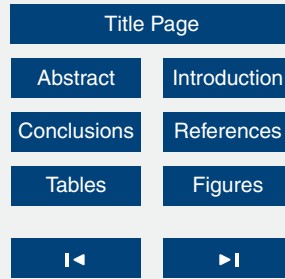

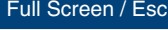

opaque or cloud emissivity if clouds are homogeneous in the FOV. The cloud top pressure (CTP) can be estimated from the calculations of this equation at each level of atmosphere. If clouds are detected, then ECA is calculated from the window channel data using the relation of

$$\alpha\epsilon_\lambda = \frac{R_\lambda - R_\lambda^{\mathrm{clr}}}{R_\lambda^{\mathrm{bcd}} - R_\lambda^{\mathrm{clr}}}, \tag{2}$$

where $R_\lambda^{\mathrm{bcd}}$ is the radiance if dense clouds are in the IFOV homogeneously. If clouds exist in the IFOV homogeneously, then the optical thickness of the clouds is represented as

$$\tau_\lambda = -\cos\theta\ln(1 - \epsilon_\lambda), \tag{3}$$

where $\theta$ is the zenith angle of the observation.

   According to several previous studies (i.e., Wylie and Menzel, 1989; Zhang and Menzel, 2002; Chang et al., 2010; Wylie et al., 2007), the $CO_2$ slicing method can estimate the cloud top height (CTH) of clouds higher than 600 hPa pressure level (corresponding to approximately 4 km) and their optical thickness greater than 0.1. On the other hand, clouds lower than 600 hPa have been discriminated with the difference between cloudy and clear sky radiances. However, this technique also only slightly detects clouds accurately because of the small contrast of atmospheric temperature between the cloud top and that of the surface.

   For the slicing method, it is assumed that cloud emissivity is equal in both bands and that clouds are infinitesimally thin. Although the errors associated with assumption of constant emissivity for two channels are negligible (Menzel et al., 1992), the latter assumption can influence CTH such that is estimated as lower than its actual height (Wielicki and Coakley, 1981). Wylie and Menzel (1989) and Hawkinson et al. (2005) also reported that $CO_2$ slicing tends to underestimate CTHs compared with lidar observations.

**AMTD**

doi:10.5194/amt-2015-371

**A development of cloud top height retrieval**

Y. Someya et al.

Discussion Paper | Discussion Paper | Discussion Paper | Discussion Paper |

## 3.2 Channel reconstruction

The pair of channels used in the slicing method is selected based on the profiles of sensitivity, which is called the weighting function and which is defined as the altitude derivation of transmittance at each channel. Several channels of Band 4 can have the weighting function peak in the same height in any wavenumber range because the wavenumber resolution of GOSAT data is much higher than that of the sensors used in previous studies using the $CO_2$ slicing method. To improve the detection accuracy, the channels were reconstructed based on the weighting function peak height. Also, the sets of the original channels were redefined as "pseudo-channels" in this study to reduce the effects of spectral random errors compared with single channel use.

The spectral range of 700–755 cm$^{-1}$ is used in the analysis. Pseudo-channels were constructed from the range of 740–755 cm$^{-1}$ for low-level cloud detection, and 700–750 cm$^{-1}$ for middle and high-level cloud detection. In these wavenumber ranges, the weighting functions and their peak height are calculated, and pseudo-channels are redefined for each 0.5 km as the sets of the original channels to have the weighting function peak within the same height ranges. Figure 1 presents an example of the channel reconstruction procedure. For the original channels that have transmittance in panel a, weighting function peak heights are calculated as panel b and the channels are sorted as panel c based on the weighting function peak heights. $X$ axis of panels c and d denote the number of channels in order of increasing the peaks of the weighting functions. The pseudo-channels are defined as the sets of the original channels in the same height range within 0.5 km for each height as panel d. The length of the bars along $x$ axis represents the number of original channels within each pseudo-channel and that corresponds to the right figure of Fig. 2. Figure 2 portrays the weighting function profiles of pseudo-channels and the number of original channels within the pseudo-channels.

Discussion Paper | Discussion Paper | Discussion Paper | Discussion Paper | Discussion Paper |

**AMTD**

doi:10.5194/amt-2015-371

**A development of cloud top height retrieval**

Y. Someya et al.

## 3.3 Channel optimization

Once the channels are prepared, the pair must be selected from pseudo-channels for the $CO_2$ slicing method calculation. The pairs were optimized from simulation studies with Pstar3 for several typical temperature profiles.

Averaged temperature profiles were calculated for each 5 K at 500 hPa in northern high latitudes (60–90° N), northern middle latitudes (30–60° N), low latitudes (30° S–30° N), southern middle latitudes (60–30° S), and southern high latitudes (90–60° S) from the atmospheric profiles at the observation points of GOSAT based on GSM-GPV data. Theoretical cloudy sky radiances observed from space for several cloud patterns were calculated using LBLRTM and Pstar3 for all of these temperature profiles. These cloudy sky radiance spectra were analyzed by the $CO_2$ slicing method algorithm. The errors of estimated CTHs from the assumed CTHs in Pstar3 were investigated for all pairs. The CTHs of low, middle, and high-level clouds were defined respectively as 1–3, 3–6, and 6–15 km. The cloud optical thickness (COT) was defined in the range of 0.05–3.0. For each level, the channel pairs for which the standard deviation of the estimated and assumed CTH is the minimum are chosen as the optimal pairs. This error analyses were conducted to select the pairs of pseudo-channels for all prepared temperature profiles. For low cloud detection, the common pair of pseudo-channels was used for observed data analysis because the detection accuracy has few differences among all temperature profiles. Although the view angle for cross track observations is about 30° maximum, the detection accuracy was almost identical to that of the nadir observation for the simultaneous studies. Therefore, both channel selection and reconstruction processes were performed without consideration of cross-track angles.

Figure 3 represents the examples of detection accuracy with randomly biased spectra using the original channel pairs and pseudo channel pairs. The biases were randomly added to simulated spectra for each original channel within 0.5 K maximum. Root mean square errors (RMSEs) between assumed and retrieved CTH are shown in color. Black grid shows CTH was not appropriately detectable more than one analysis. Many

**AMTD**

doi:10.5194/amt-2015-371

**A development of cloud top height retrieval**

Y. Someya et al.

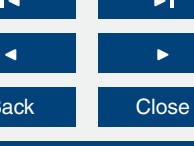

grids of left figure using original channel pairs are filled in black. On the other hand, a lot of grids of right figure using reconstructed channel pairs are colored and RMSEs are generally smaller than left figure. This fact shows that channel reconstruction is useful to reduce the effect of random biases in observed spectra.

## 3.4 Application to GOSAT data

For analysis of the observed radiance spectra, the pair of the pseudo-channels is determined based on results of the error investigations presented in Sect. 3.3, referring to latitude and temperature at 500 hPa for each observation. In Eq. (1), $R_\lambda$, and $R_\lambda^{clr}$ respectively denote the observed radiance and the calculated clear sky radiance. The right side of Eq. (1) is calculated from temperature and water vapor profiles derived from GPV datasets and optical thickness of layers calculated using LBLRTM. Although the surface skin temperature is required for analysis, it is not included in GPV data over land. Therefore, the air temperature at 2 m height above the surface was assumed as the surface skin temperature over land. However, they are generally not consistent because of surface heating caused by solar radiation or radiative cooling during nighttime. Consequently, these differences can be the main cause of detection errors. Over the ocean, SST included in GPV datasets was used as the surface skin temperature. Clear sky radiance calculations were made by LBLRTM considering gas absorption and Rayleigh scattering by molecules, but not scattering by particles such as clouds and aerosols. This calculation also considers the angle of the sensor's line of sight. Spectral data from FTS present the problem that the wavelength and sensitive height of channels shifts slightly day by day because the laser system misalignment gradually occurred on orbit (Kuze et al., 2012). The channel re-construction procedure proposed herein can reduce the effect through averaging of the shifts of wavelength positions of each spectral channel.

Slicing calculations were performed up to three times with different channel pairs for high, middle, and low level altitudes such as the "top-down approach" presented in Menzel et al. (2008). A clear scene was identified by the conditions under which the

**AMTD**

doi:10.5194/amt-2015-371

**A development of cloud top height retrieval**

Y. Someya et al.

brightness temperature difference in the most transmissive channel between calculated and observed radiance was less than the measurement accuracy (0.5 K), the observed brightness temperature was more than 10 K higher than the calculated brightness temperature, or the slicing method detected lowest layer. If clouds with respective CTH and COT of 10 km and 0.02 are assumed with the temperature profile in the Northern Hemisphere, the brightness temperature difference is almost 0.5 K with the most transmissive channel. In this case, clouds were detectable with accuracy within 2 km. If clouds are not detected and it is not identified as clear, the seen is flagged as "uncertain".

## 4 Validation of the algorithm using CALIPSO data

In this section, the results from the improved slicing were compared mainly with those from CALIPSO observations. In addition, they were also compared the results from TIR threshold technique and CAI in Sect. 4.1 and HIRS-like slicing in Sect. 4.2. TIR threshold technique is based on spectral brightness temperature differences and is currently applied to operational Level 2 processing. The HIRS-like slicing is the traditional $CO_2$ slicing with the original GOSAT channels corresponding to HIRS sensor (Wylie et al., 1994).

### 4.1 Statistical comparisons

Because the orbital paths of GOSAT and CALIPSO are not synchronized frequently, only a few co-located observations can be done, thus the latitudinal areas of co-locations are restricted, as shown in Sect. 4.2. Therefore, we examined monthly averaged results on a global scale to elucidate the consistency of regional and seasonal variability of cloud data retrieved using the slicing method proposed in this study. All the data for GOSAT and CALIPSO observed in January and July were analyzed. The results were averaged for each month and validated statistically using those from

**AMTD**

doi:10.5194/amt-2015-371

**A development of cloud top height retrieval**

Y. Someya et al.

**AMTD**

doi:10.5194/amt-2015-371

**A development of cloud top height retrieval**

Y. Someya et al.

CALIPSO observations. The slicing does not provide information about lower layers of the cloud top, so only the uppermost cloud layer data of CALIPSO were used for comparison. The total cloud amount (CA), high-level cloud amount (CAH), middle-level cloud amount (CAM), and low-level cloud amount (CAL) are defined respectively as the ratios of the numbers of the observations for which the total, high-level ($CTP < 440\,hPa$), middle-level ($440 \leq CTP < 680\,hPa$), and low-level ($CTP \geq 680\,hPa$) clouds are detected to the number of total observations. In addition, the relative high-level cloud amount (CAHR), relative middle-level cloud amount (CAMR), and relative low-level cloud amount (CALR) are scaled respectively as CAHR = CAH/CA, CAMR = CAM/CA, and CALR = CAL/CA. Table 1 presents the monthly mean values of CA, CAHR, CAMR, and CALR from the slicing method and CALIPSO over all surfaces, over the ocean, and over land in January and July. The GOSAT values in this table generally agreed well with CALIPSO data, except for some points such as that CALR over lands from the slicing is higher than those from CALIPSO.

### 4.1.1 Latitudinal distribution

Figure 4 presents latitudinal variations of the monthly mean of CA during daytime and nighttime retrieved using four methods: the improved slicing method, TIR threshold technique, CALIPSO, and CAI. The values from CAI are described only in the daytime panel because CAI data are not obtainable during nighttime. All observations show similar trends: CA is high in the Tropics, low in the subtropical high-pressure belt, and increasing with latitude at middle and high latitudes. The highest value of CA was shown by CAI because the size of IFOV of FTS is higher than CALIOP and CAI can determine very small clouds in the IFOV. TIR threshold showed lowest values because optically thin clouds or partially clouds are detected only slightly using this technique. The values from the slicing are closer to those from CALIPSO than either the CAI or TIR threshold, but they appear to be somewhat high in Northern Hemisphere during nighttime in January and Southern subtropics in July.

Figure 5 shows the latitudinal variations of monthly mean CAHR, CAMR, and CALR for all the day. Furthermore, in this figure, the distribution trends from slicing are similar to those from CALIPSO for all altitudes. In this figure, CAL from slicing is higher than that from CALIPSO especially in high latitudes of winter hemisphere and this engender to high CA during nighttime in Fig. 4. For the analysis, the surface skin temperature is assumed to be equal to the air temperature at 2 m height above the surface, as noted in Sect. 2. However, land surface temperature (LST) becomes lower than surface air temperature by radiative cooling in nighttime. The fact of higher CA especially during nighttime in Northern Hemisphere winter implies that these biases are caused by low surface temperature over land. Table 1 shows that CA from slicing over the land is higher than that over the ocean in contrast those from CALIPSO. This is possibly related to results from CAL. If surface temperature biases exist, then clear sky determination using brightness temperature differences is ineffective. Moreover, it can be difficult to detect a surface by the slicing. In addition, a tendency by which CAH decreases and CAL increases with latitude in these areas is presented in Fig. 5. Menzel et al. (1992) reported that the estimated CTH by the slicing tends to be too high (low) if the actual surface skin temperature is higher (lower) than assumed. This surface skin temperature bias is probably greater at high latitudes because the nighttime is longer in winter. Because of the longer nighttime, high-level clouds can be detected as lower-level clouds by the slicing.

In Fig. 5, CAHR from the slicing is lower than that from CALIPSO in the Tropics, especially in January, however, CALR is higher in contrast. This underestimation of CTH is explainable mainly by two causes: optically very thin cirrus and multilayer structure of clouds. CALIPSO can detect optically very thin clouds with optical thickness of approximately 0.02 (Winker et al., 2007). However, the slicing can reportedly detect clouds in optical thickness down to 0.1 (Wylie and Manzel, 1999). Optically very thin clouds such as subvisible cirrus with optical thickness of approximately 0.03 are known to occur frequently in the Tropics, especially during boreal winter. Their annual mean occurrences are approximately 0.1 during daytime and 0.15 during nighttime (Sassen et al., 2009;

**AMTD**

doi:10.5194/amt-2015-371

**A development of cloud top height retrieval**

Y. Someya et al.

Discussion Paper | Discussion Paper | Discussion Paper | Discussion Paper

Martins et al., 2011). Furthermore, clouds in the Tropics often have a multilayer structure (Wu et al., 2011). Menzel et al. (1992) described that the estimation error in height is greatest when transmissive clouds exist near the tropopause over opaque clouds in the middle troposphere. In this situation, slicing may underestimate the top of upper-level clouds same as negative surface temperature biases. Hence, the reason of low CAHR and high CALR in the Tropics is probably because the slicing underestimated the upper-level cloud top or detected underlying clouds in the situations of multilayer clouds and optically very thin clouds.

### 4.1.2 Vertical distribution

Figure 6 presents vertical frequency distributions of CTH from the slicing and CALIPO. Although clouds higher than 15 km and lower than 1 km are underestimated and those lower than 10 km are overestimated, the distributions generally agree. As described in Sect. 4.1.1, CTH was underestimated by slicing because of very thin cirrus near the tropopause or cloud multilayer structures. The extremely low-level clouds, which have their tops below 1 km, are detected only to a slight degree. Overall, the slicing seems to exhibit a tendency of slight overestimation of CTH of low-level clouds.

The zonally averaged vertical distributions of monthly mean cloud top occurrence from the slicing and CALIPSO are portrayed in Fig. 7. Characteristics are readily apparent, such as high frequency above 10 km in the Inter Tropical Convergence Zone (ITCZ), decreasing CTH with latitude, and low-level clouds are frequently detected at middle and high latitudes. However, the maximum value of high-level clouds from the slicing is lower than that from CALIPSO because of the causes described above. In the Tropics, the level of the maximum value from the slicing is lower than that from CALIPSO, which means that the slicing had underestimated CTH. For optically thin clouds, it is expected that the height level of clouds detected by the slicing fall below CALIPSO CTH. This phenomenon is described in other reports of studies about cloud retrieval using nadir-looking passive sensors (e.g. Wu et al., 2009). In addition, slicing

Discussion Paper | Discussion Paper | Discussion Paper | Discussion Paper

**AMTD**

doi:10.5194/amt-2015-371

**A development of cloud top height retrieval**

Y. Someya et al.

only slightly detected low-level cloud tops lower than 1 km at middle and high latitudes, where high occurrences are observed from CALIPSO.

### 4.1.3   Horizontal distribution

Figures 8 and 9 present the horizontal distributions of monthly mean CA, CAH, CAM, and CAL from the slicing, CALIPSO, and their differences within 2.5° × 2.5° horizontal grids. A map showing cloud amounts between the slicing and CALIPSO show similarity and generally agreement of characteristics such as high amounts in ITCZ and the Western Pacific warm pool, as reported in previous studies (e.g. WCRP, 2012). However, slicing results show some difference from CALIPSO as follows. In the Tropics, CAH is underestimated. This area corresponds to the area in which sub-visible cirrus and multilayer clouds frequently occur (Sassen et al., 2009; Martins et al., 2011; Wu et al., 2011). These cause detection error or a sensitivity difference of sensors, probably engendering CAH difference.

CAL is overestimated over land at high latitudes. This is probably because of surface temperature biases described in Sect. 4.1.1. However, the large underestimation is apparent for CAL on the west coasts of continents, especially California in July. There, SST is lower than in other areas because of the cold current and upwelling of ocean water and downward air flow generally occurs because of subtropical high-pressure belts. Therefore, strong inversion layers develop frequently, which is a good condition for occurrences of marine stratocumulus with cloud tops as high as 2 km. As Figs. 6 and 7 show, the slicing only slightly detects low clouds which have cloud tops of less than 1 km because of the small contrast of temperatures between the surface and clouds. In addition, the occurrence of an inversion layer is a major cause of detection error because slicing uses the vertical temperature gradient. In the Southern Hemisphere, slicing tends to overestimate CAL. However, low-level partially clouds are frequently occur in this area, thus this overestimation is explainable from considering the difference of size of IFOV between GOSAT and CALIPSO.

Discussion Paper | Discussion Paper | Discussion Paper | Discussion Paper | Discussion Paper |

**AMTD**

doi:10.5194/amt-2015-371

**A development of cloud top height retrieval**

Y. Someya et al.

## 4.2 Comparison for coincident observations

In this section, the properties derived from slicing are compared with those from co-located CALISPO observations within 5 km and 2 min, although such data are not so numerous. Actually, 123 of GOSAT observations and 316 of CALIPSO observations were found in 2010 only at the middle latitudes of 24.4–56.6° N. The geographical locations of the observations were presented in Fig. 10. The reason for the number of the data is the difference of the size of IFOV and the spatial intervals of the footprint. The latitudinal limitation is attributable to their different orbital paths and revisit cycles, as described in Sect. 2. Figure 11 presents a comparison of CTH derived from the slicing and CALIPSO. Optical thicknesses of uppermost clouds from CALIPSO are shown in color. The left panel shows results obtained using the traditional slicing method with original channels corresponding to the HIRS sensor (Wylie et al., 1994). The slicing calculations were performed three times in maximum as top-down approach shown in Menzel et al. (2008). The right panel shows results obtained using the improved slicing method developed in this study. This figure shows that improved slicing is more accurate and this means that channel reconstruction and optimization is effective to retrieve CTH. Especially, it enables detection of lower-level clouds down to approximately 1.5 km. Holz et al. (2006) noted that the detection of low-level clouds below 3 km is challenging for IR measurements. Thus our results mean that the improvement expands the detectable CTH. In some cases, the slicing detected low-level clouds in spite that CALIPSO detected as high-level clouds in this figure. Investigating the observed data from CAI and CALIPSO in those four extreme cases, the clouds in the IFOV have multi-layer structure in two cases and are optically thin in the other two cases. This fact shows that these causes sometimes occur underestimation of CTH by the improved slicing.

# AMTD

doi:10.5194/amt-2015-371

**A development of cloud top height retrieval**

Y. Someya et al.

## 5 Discussions and conclusions

The cloud detection algorithm based on a cirrus detection technique, $CO_2$ slicing method, was developed for high-resolution TIR spectral data with channel reconstruction and channel optimization. Based on the weighting function, where peak heights correspond to the most sensitive height of the channels, pseudo-channels were redefined as sets of original channels in the same height range for each 0.5 km height increment to decrease the effects of random errors of observed spectra. Pairs of these pseudo-channels for use in slicing calculations were optimized for several typical temperature profiles as indicators of latitude and temperature at 500 hPa based on simulation studies with Pstar3. The simultaneous studies showed that these improvements reduce the effects of random errors of spectra. For GOSAT data analysis, optimal pairs of pseudo-channels were chosen from the indicators for each observation.

The improved slicing algorithm was demonstrated using TIR spectra data in 2010 observed by TANSO-FTS/GOSAT. Then the analysis results were validated using CALIPSO observation data. Statistical comparison showed that the analyzed results from the slicing generally agreed with those from CALIPSO, although some differences are apparent. Slicing tends to underestimate CAH and detect as lower CTH near the tropopause in the Tropics, probably because of the optically very thin clouds and multi-layer structure of clouds in this region. CAL is overestimated over land in high latitudes during winter as results that low-level clouds were detected in the clear sky seen or that high-level clouds are detected as lower-level clouds probably because of very cold surface. Marine stratocumulus clouds on the west coast of continents are also less detected must likely because of their extremely low CTH and the occurrences of temperature inversion layer. Compared to CAI and TIR threshold techniques, slicing represents the closest latitudinal variations of CA to CALIPSO. This close variation implies that this algorithm can improve cloud screening of GOSAT, which leads to improvement of gas retrieval, especially with TIR.

To investigate the accuracy of the algorithm more quantitatively, CTHs from the slicing were compared with those from co-located CALIPSO observations. Comparison of coincident observations obtained from GOSAT and CALIPSO revealed that the improved slicing has higher accuracy than those of the HIRS-like slicing method. Our results demonstrated that low-level cloud tops as low as approximately 1.5 km are detectable using the method demonstrated in this study. However, in the situation of multilayer structure or optically thin clouds, CTH were sometimes underestimated from the improved slicing.

TANSO-FTS TIR spectra have some biases. The effects of random errors were decreased by channel reconstruction as demonstrated in Fig. 3, however, systematic biases can also affect to detection accuracy. The simultaneous studies showed that the slicing estimates CTH lower (higher) especially for optically thick clouds if the negative (positive) systematic biases are included for entire channels (not shown in figure). Gero et al. (2014) reported that TIR spectra from GOSAT have slight bias compared with the Atmospheric Infrared Sounder (AIRS) and the infrared atmospheric sounding interferometer (IASI). Especially, the comparison with IASI in the cold region had shown large errors. These biases cannot be removable by our improvements. Therefore, it is possible that the overestimation of CA in the high latitudes is partially resulted by these biases.

Comparison results show that the algorithm developed for this study has high detectability of clouds, approximating that of CALIPSO. Therefore, it can be expected that the accuracy of cloud screening and gas retrievals from GOSAT data would be improved if it were used. Application of this algorithm is planned for data from the GOSAT-2 satellite which is scheduled to be launched in 2018.

*Acknowledgements.* This study was partially supported as a program of Ministry of Education, Culture, Sports, Science and Technology – Japan (MEXT), "Green Network of Excellence – Environmental Information" (GRENE-ei).

**AMTD**

doi:10.5194/amt-2015-371

**A development of cloud top height retrieval**

Y. Someya et al.

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

## AMTD

doi:10.5194/amt-2015-371

A development of cloud top height retrieval

Y. Someya et al.



Kuze, A., Suto, H., Nakajima, M., and Hamazaki, T.: Thermal and near infrared sensor for carbon observation Fourier-transform spectrometer on the Greenhouse Gases Observing Satellite for greenhouse gases monitoring, Appl. Optics, 48, 6716–6733, doi:10.1364/ao.48.006716, 2009.

Kuze, A., Suto, H., Shiomi, K., Urabe, T., Nakajima, M., Yoshida, J., Kawashima, T., Yamamoto, Y., Kataoka, F., and Buijs, H.: Level 1 algorithms for TANSO on GOSAT: processing and on-orbit calibrations, Atmos. Meas. Tech., 5, 2447–2467, doi:10.5194/amt-5-2447-2012, 2012.

Martins, E., Noel, V., and Chepfer, H.: Properties of cirrus and subvisible cirrus from nighttime Cloud-Aerosol Lidar with Orthogonal Polarization (CALIOP), related to atmospheric dynamics and water vapor, J. Geophys. Res.-Atmos., 116, D02208, doi:10.1029/2010jd014519, 2011.

Menzel, W. P., Smith, W. L., and Stewart, T. R.: Improved cloud motion wind vector and altitude assignment using VAS, J. Clim. Appl. Meteorol., 22, 377–384, 1983.

Menzel, W. P., Wylie, D. P., and Strabala, K. I.: Seasonal and diurnal changes in cirrus clouds as seen in 4 years of observations with the VAS, J. Appl. Meteorol., 31, 370–385, 1992.

Menzel, W. P., Frey, R. A., Zhang, H., Wylie, D. P., Moeller, C. C., Holz, R. E., Maddux, B., Baum, B. A., Strabala, K. I., and Gumley, L. E.: MODIS global cloud-top pressure and amount estimation: algorithm description and results, J. Appl. Meteorol. Clim., 47, 1175–1198, 2008.

Morino, I., Uchino, O., Inoue, M., Yoshida, Y., Yokota, T., Wennberg, P. O., Toon, G. C., Wunch, D., Roehl, C. M., Notholt, J., Warneke, T., Messerschmidt, J., Griffith, D. W. T., Deutscher, N. M., Sherlock, V., Connor, B., Robinson, J., Sussmann, R., and Rettinger, M.: Preliminary validation of column-averaged volume mixing ratios of carbon dioxide and methane retrieved from GOSAT short-wavelength infrared spectra, Atmos. Meas. Tech., 4, 1061–1076, doi:10.5194/amt-4-1061-2011, 2011.

Ota, Y., Higurashi, A., Nakajima, T., and Yokota, T.: Matrix formulations of radiative transfer including the polarization effect in a coupled atmosphere–ocean system, J. Quant. Spectrosc. Ra., 111, 878–894, 2010.

Rothman, L. S., Jacquemart, D., Barbe, A., Benner, D. C., Birk, M., Brown, L. R., Carleer, M. R., Chackerian, C., Chance, K., Coudert, L. H., Dana, V., Devi, V. M., Flaud, J. M., Gamache, R. R., Goldman, A., Hartmann, J. M., Jucks, K. W., Maki, A. G., Mandin, J. Y., Massie, S. T., Orphal, J., Perrin, A., Rinsland, C. P., Smith, M. A. H., Tennyson, J., Tolchenov, R. N., Toth, R. A., Vander Auwera, J., Varanasi, P., and Wagner, G.: The HI-

Discussion Paper | Discussion Paper | Discussion Paper | Discussion Paper

**AMTD**

doi:10.5194/amt-2015-371

**A development of cloud top height retrieval**

Y. Someya et al.

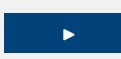

TRAN 2004 molecular spectroscopic database, J. Quant. Spectrosc. Ra., 96, 139–204, doi:10.1016/j.jqsrt.2004.10.008, 2005.

Saitoh, N., Imasu, R., Ota, Y., and Niwa, Y.: $CO_2$ retrieval algorithm for the thermal infrared spectra of the Greenhouse Gases Observing Satellite: potential of retrieving $CO_2$ vertical profile from high-resolution FTS sensor, J. Geophys. Res.-Atmos., 114, D17305, doi:10.1029/2008jd011500, 2009.

Sassen, K., Wang, Z., and Liu, D.: Cirrus clouds and deep convection in the tropics: insights from CALIPSO and CloudSat, J. Geophys. Res.-Atmos., 114, D00H06, doi:10.1029/2009jd011916, 2009.

Smith, W. L. and Platt, C. M. R.: Comparison of satellite-deduced cloud heights with indications from radiosonde and ground-based laser measurements, J. Appl. Meteorol., 17, 1796–1802, 1978.

Uchino, O., Kikuchi, N., Sakai, T., Morino, I., Yoshida, Y., Nagai, T., Shimizu, A., Shibata, T., Yamazaki, A., Uchiyama, A., Kikuchi, N., Oshchepkov, S., Bril, A., and Yokota, T.: Influence of aerosols and thin cirrus clouds on the GOSAT-observed $CO_2$: a case study over Tsukuba, Atmos. Chem. Phys., 12, 3393–3404, doi:10.5194/acp-12-3393-2012, 2012.

Wielicki, B. A. and Coakley, J. A.: Cloud retrieval using infrared sounder data – error analysis, J. Appl. Meteorol., 20, 157–169, 1981.

Winker, D. M., Hunt, W. H., and McGill, M. J.: Initial performance assessment of CALIOP, Geophys. Res. Lett., 34, L19803, doi:10.1029/2007gl030135, 2007.

Wu, D., Hu, Y. X., McCormick, M. P., and Yan, F. Q.: Global cloud-layer distribution statistics from 1 year CALIPSO lidar observations, Int. J. Remote Sens., 32, 1269–1288, 2011.

Wu, D. L., Ackerman, S. A., Davies, R., Diner, D. J., Garay, M. J., Kahn, B. H., Maddux, B. C., Moroney, C. M., Stephens, G. L., Veefkind, J. P., and Vaughan, M. A.: Vertical distributions and relationships of cloud occurrence frequency as observed by MISR, AIRS, MODIS, OMI, CALIPSO, and CloudSat, Geophys. Res. Lett., 36, 5, doi:10.1029/2009gl037464, 2009.

Wunch, D., Toon, G. C., Blavier, J.-F. L., Washenfelder, R. A., Notholt, J., Connor, B. J., Griffith, D. W. T., Sherlock, V., and Wennberg, P. O.: The total carbon column observing network, Philos. T. R. Soc. A, 369, 2087–2112, 2011.

Wylie, D., Jackson, D. L., Menzel, W. P., and Bates, J. J.: Trends in global cloud cover in two decades of HIRS observations, J. Climate, 18, 3021–3031, 2005.

Wylie, D., Eloranta, E., Spinhirne, J. D., and Palm, S. P.: Comparison of cloud cover statistics from the GLAS lidar with HIRS, J. Climate, 20, 4968–4981, doi:10.1175/jcli4269.1, 2007.

**AMTD**

doi:10.5194/amt-2015-371

**A development of cloud top height retrieval**

Y. Someya et al.

Discussion Paper | Discussion Paper | Discussion Paper | Discussion Paper

Wylie, D. P. and Menzel, W. P.: Two years of cloud cover statistics using VAS, J. Climate, 2, 380–392, 1989.

Wylie, D. P. and Menzel, W. P.: Eight years of high cloud statistics using HIRS, J. Climate, 12, 170–184, 1999.

Wylie, D. P., Menzel, W. P., Woolf, H. M., and Strabala, K. I.: 4 years of global cirrus cloud statistics using HIRS, J. Climate, 7, 1972–1986, 1994.

Yoshida, Y., Kikuchi, N., Morino, I., Uchino, O., Oshchepkov, S., Bril, A., Saeki, T., Schutgens, N., Toon, G. C., Wunch, D., Roehl, C. M., Wennberg, P. O., Griffith, D. W. T., Deutscher, N. M., Warneke, T., Notholt, J., Robinson, J., Sherlock, V., Connor, B., Rettinger, M., Sussmann, R.,

Ahonen, P., Heikkinen, P., Kyrö, E., Mendonca, J., Strong, K., Hase, F., Dohe, S., and Yokota, T.: Improvement of the retrieval algorithm for GOSAT SWIR $XCO_2$ and $XCH_4$ and their validation using TCCON data, Atmos. Meas. Tech., 6, 1533–1547, doi:10.5194/amt-6-1533-2013, 2013.

Zhang, H. and Menzel, W. P.: Improvement in thin cirrus retrievals using an emissivity-adjusted

$CO_2$ slicing algorithm, J. Geophys. Res.-Atmos., 107, 4327, doi:10.1029/2001jd001037, 2002.

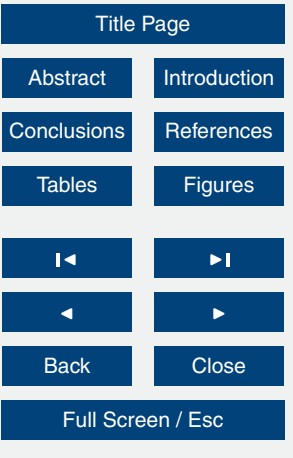

**AMTD**

doi:10.5194/amt-2015-371

**A development of cloud top height retrieval**

Y. Someya et al.

Discussion Paper | Discussion Paper | Discussion Paper | Discussion Paper

**AMTD**

doi:10.5194/amt-2015-371

**A development of cloud top height retrieval**

Y. Someya et al.

**Table 1.** CA, CAH, CAM, and CAL from the slicing and CALIPSO over all surfaces, ocean, and land in January and July.

|     |       |         | CA   | CAH (%) | CAM (%) | CAL (%) |
|-----|-------|---------|------|---------|---------|---------|
| Jan | All   | Slicing | 0.74 | 45      | 16      | 39      |
|     |       | CALIPSO | 0.68 | 60      | 12      | 28      |
|     | Ocean | Slicing | 0.73 | 43      | 17      | 40      |
|     |       | CALIPSO | 0.70 | 56      | 11      | 33      |
|     | Land  | Slicing | 0.75 | 49      | 16      | 35      |
|     |       | CALIPSO | 0.64 | 70      | 14      | 16      |
| Jul | All   | Slicing | 0.69 | 51      | 13      | 36      |
|     |       | CALIPSO | 0.66 | 52      | 19      | 29      |
|     | Ocean | Slicing | 0.69 | 46      | 13      | 41      |
|     |       | CALIPSO | 0.68 | 57      | 9       | 34      |
|     | Land  | Slicing | 0.70 | 65      | 11      | 23      |
|     |       | CALIPSO | 0.61 | 66      | 19      | 14      |

AMTD

doi:10.5194/amt-2015-371

**A development of cloud top height retrieval**

Y. Someya et al.

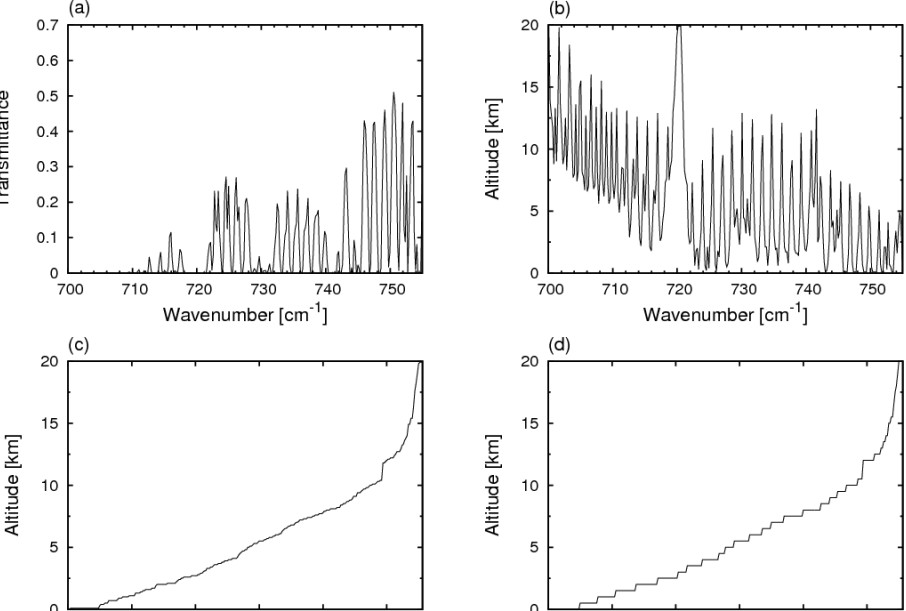

**Figure 1. (a)** Integrated transmittance at each channel of FTS in the wavenumber region of 700–755 cm$^{-1}$. **(b)** Calculated weighting function peak height at each channel of FTS in the same region of **(a)**. **(c)** Channels are sorted based on their weighting function peak heights. **(d)** Channels for which weighting function peak heights are in the same height range are redefined as pseudo-channels for each 0.5 km.



**AMTD**

doi:10.5194/amt-2015-371

**A development of cloud top height retrieval**

Y. Someya et al.

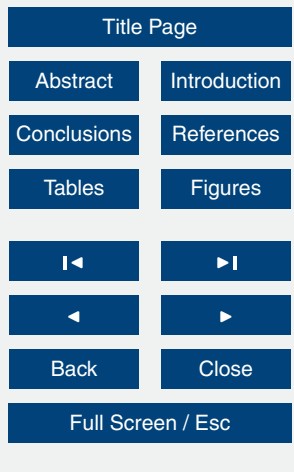

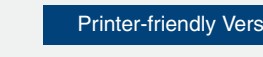



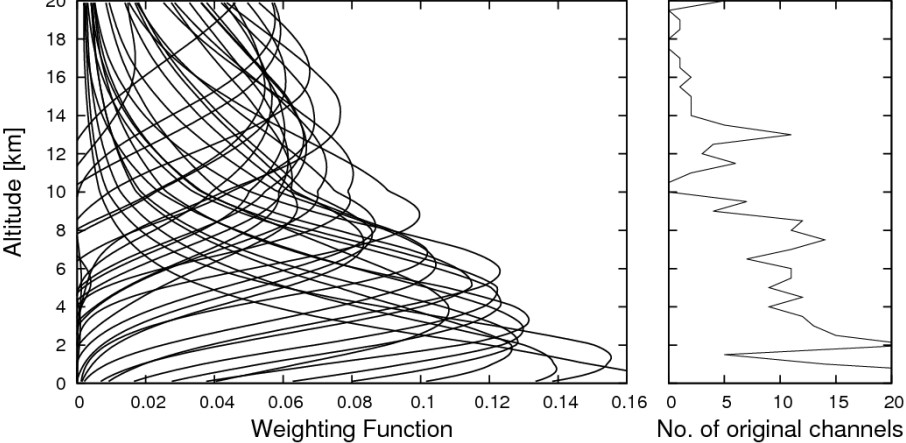

**Figure 2.** Weighting function profiles of reconstructed channels using the mid-latitude summer (MLS) atmospheric profile model of Air Force Geophysics Laboratory (AFGL) (left) and the number of original channels within each reconstructed channel (right).

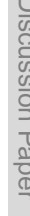

# AMTD

doi:10.5194/amt-2015-371

**A development of cloud top height retrieval**

Y. Someya et al.

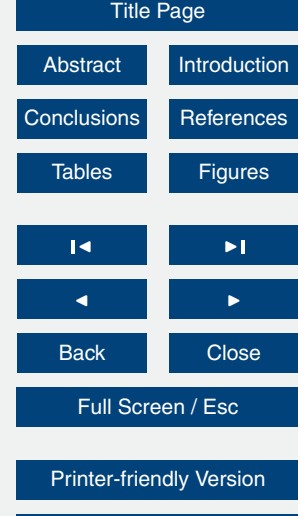

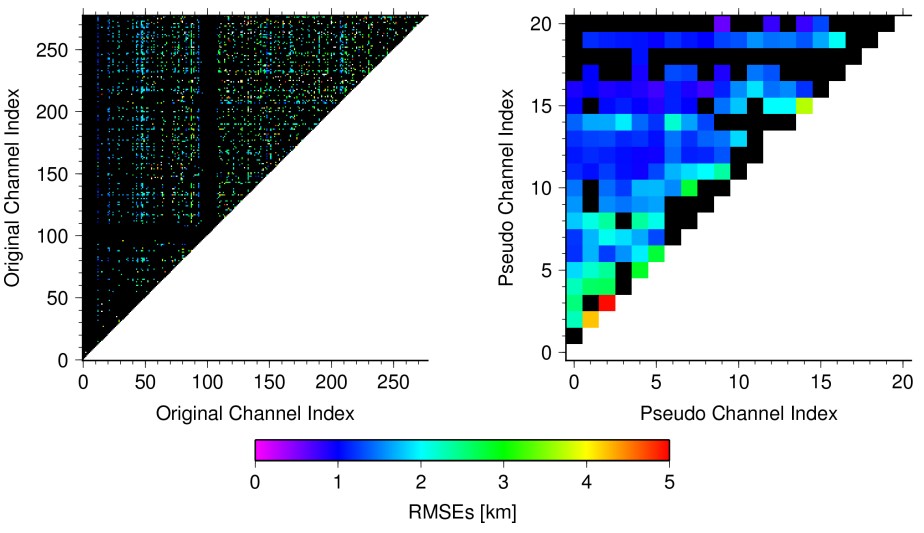

**Figure 3.** The examples of the detection accuracy with spectral biases using the original channel pairs between 700–755 cm$^{-1}$ (left) and the reconstructed channel pairs (right) based on the simultaneous studies with Pstar3. The axes of left figure correspond to original channels from 700 to 755 cm$^{-1}$ and those of right figure correspond to the weighting function peak heights of pseudo channels from 0 to 10 km. The clouds with CTH of 9, 12, 15 km and COT of 0.05–5.0 were assumed for each simulations. Tropical atmospheric profile model of AFGL was used. The spectral biases were randomly added to the radiances for each original channel within ±0.5 K in maximum. RMSEs of retrieved CTH were presented in color. If CTH was not appropriately detectable more than one analysis, the grid is filled with black color.

Discussion Paper | Discussion Paper | Discussion Paper | Discussion Paper

**AMTD**

doi:10.5194/amt-2015-371

**A development of cloud top height retrieval**

Y. Someya et al.

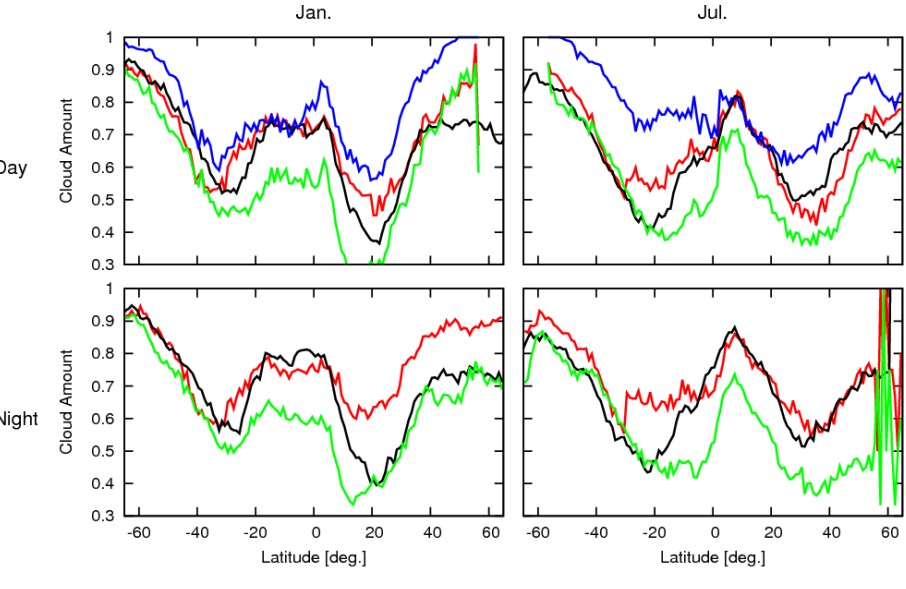

**Figure 4.** Latitudinal variations of monthly mean CA during daytime and nighttime in January and July from the slicing (red line), CALIPSO (black line), CAI (blue line), and the TIR threshold method (green line).

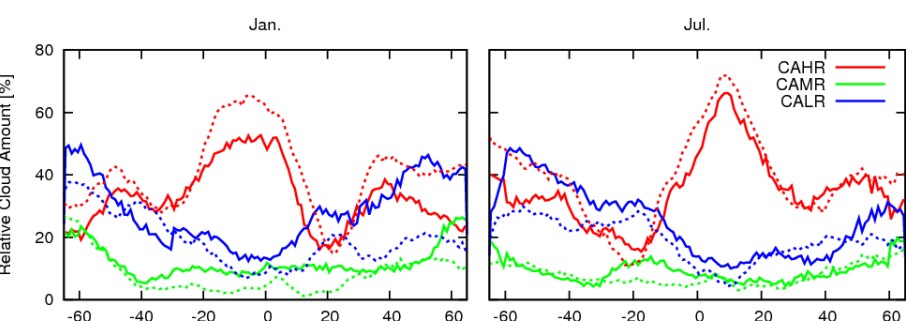

**Figure 5.** Latitudinal variations of monthly mean CAHR (red line), CAMR (green line), and CALR (blue line) from the slicing (solid line) and CALIPSO (dashed line) in January and July.

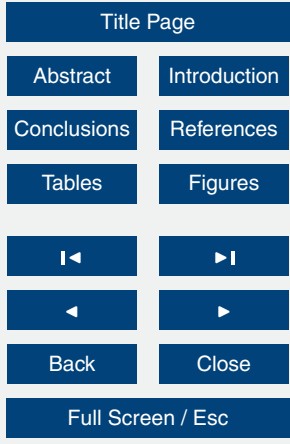

**AMTD**

doi:10.5194/amt-2015-371

**A development of cloud top height retrieval**

Y. Someya et al.

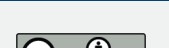

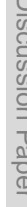
Discussion Paper | Discussion Paper | Discussion Paper | Discussion Paper

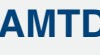

**AMTD**

doi:10.5194/amt-2015-371

**A development of cloud top height retrieval**

Y. Someya et al.

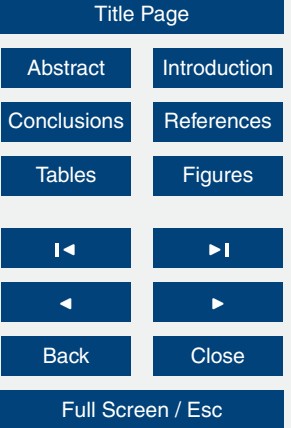

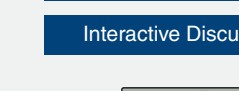

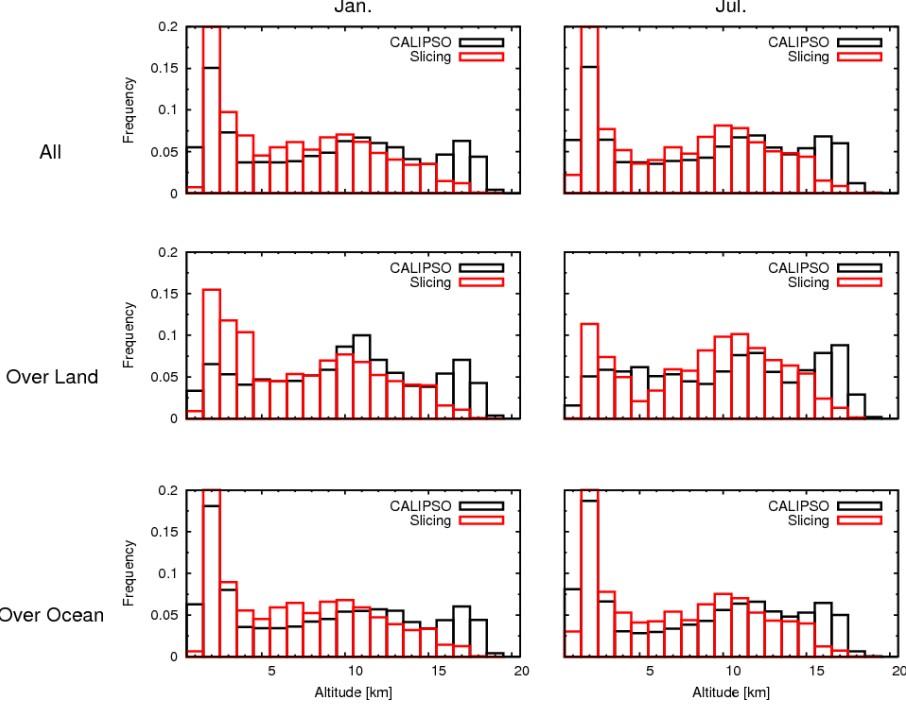

**Figure 6.** Vertical frequency distributions of CTHs from the slicing (red bars) and CALIPSO (black bars) over all surfaces (top), over land (middle), and over the ocean (bottom) in January (left column) and July (right column).

Discussion Paper | Discussion Paper | Discussion Paper | Discussion Paper

**AMTD**

doi:10.5194/amt-2015-371

**A development of cloud top height retrieval**

Y. Someya et al.



**Figure 7.** Zonally averaged vertical distributions of monthly mean uppermost cloud top occurrence obtained from the slicing (left column) and CALIPSO (right column) in January (top panel) and July (bottom panel). Zonally averaged values are shown within grid size of 5° horizontally and 1 km vertically.

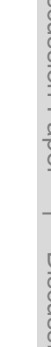



# AMTD

doi:10.5194/amt-2015-371

## A development of cloud top height retrieval

Y. Someya et al.

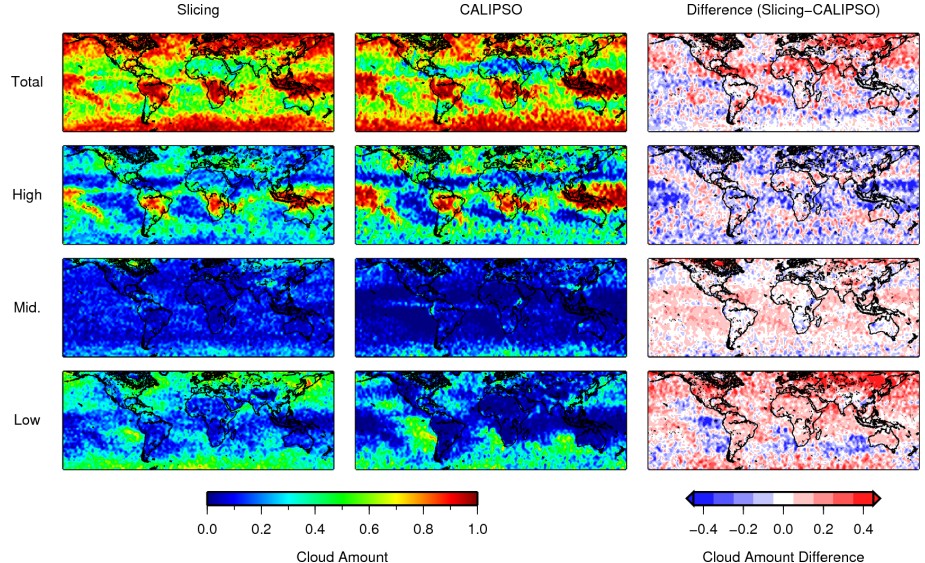

**Figure 8.** Horizontal distribution maps of monthly mean CA (top), CAH (middle top), CAM (middle bottom), and CAL (bottom) from the slicing (left column), CALIPSO (middle column), and their differences (right column) in January.

**AMTD**

doi:10.5194/amt-2015-371

**A development of cloud top height retrieval**

Y. Someya et al.

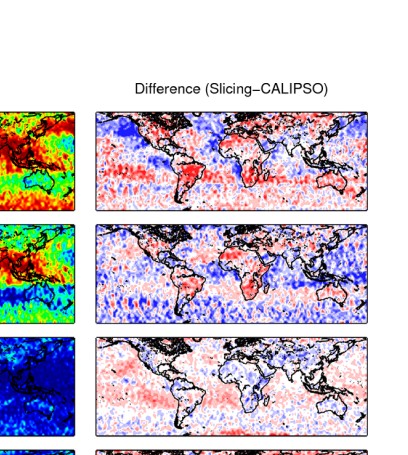

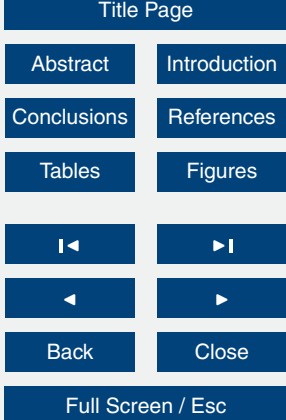

**Figure 9.** Same as in Fig. 8 but for July.

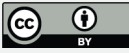

Discussion Paper | Discussion Paper | Discussion Paper | Discussion Paper | Discussion Paper

**AMTD**

doi:10.5194/amt-2015-371

**A development of cloud top height retrieval**

Y. Someya et al.

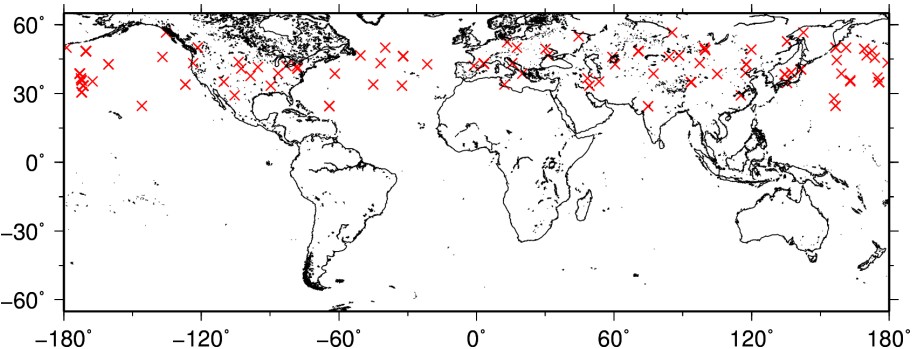

**Figure 10.** Geographical locations of coincident observations between GOSAT and CALIPSO within 5 km and 2 min in 2010.

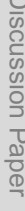

**AMTD**

doi:10.5194/amt-2015-371

**A development of cloud top height retrieval**

Y. Someya et al.

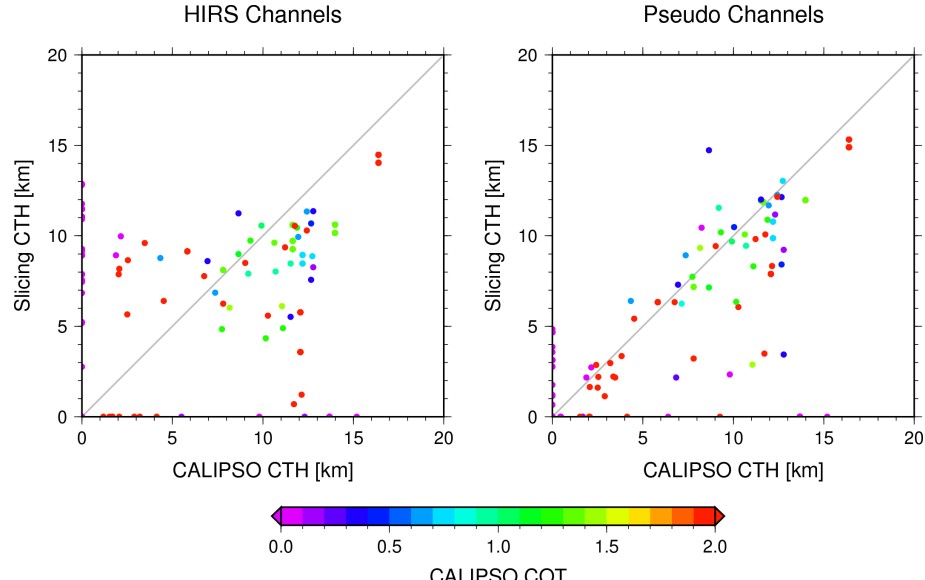

**Figure 11.** Comparison of CTH derived from CALIPSO and slicing method with the pair of original channels corresponding to the HIRS sensor (left), and obtained from the improved slicing method with the optimal pair of pseudo-channels (right). The color is optical thickness of uppermost clouds from CALIPSO.