# Peer review of "A development of cloud top height retrieval using thermal infrared spectra observed with GOSAT and comparison with CALIPSO data"

_Atmospheric Measurement Techniques, 2015_

## Referee Comment (RC1) · Anonymous Referee #1 · 29 Jan 2016

This manuscript uses CO2 slicing method on thermal infrared spectra to retrieve cloud amount and cloud height and compared with data from other satellites. The study reports a reasonable practice though the idea is not novel. But as their results show, the manuscript does show an improvement to the CO2 slicing method with spectral data. This paper is well prepared and demonstrates an useful approach to an important, though old, research topic. The data and method applied are convincing. The presentation is clear and concise. This reviewer recommends this manuscript be published after minor revisions as suggested as follows

1. When talking about CO2 slicing method, the authors should carefully introduce not only its benefit, but also its limitations in applications. E.g., can it be accurate when

[Figure]

cloud is very thin, such as optical depth smaller than ~0.3? Does background surface temperature and atmospheric temperature profile and water vapor profile result in any uncertainty to the cloud retrieval? How significantly?

2. For Eq (1), in remote sensing practice, how can clear-sky R be obtained? This may be the most significant drawback of this method. The authors should justify the treatment.

3. In Section 4.1.2. "CTH was underestimated by slicing because of very thin cirrus near ..." why?

4. CALIPSO has significant errors for detecting optically thin stuff, especially during daytime, this should be discussed in the paper.

5. When discussing about historical background information about very thin cloud detection in Introduction, the data are not complete and updated. For references, please also cite the following 2 papers for latest developments in this important field.

(1) Wenbo Sun, Rosemary R. Baize, Gorden Videen, Yongxiang Hu, and Qiang Fu, "A method to retrieve super-thin cloud optical depth over ocean background with polarized sunlight", Atmos. Chem. Phys., 15, 11909-11918, doi: 10.5194/acp-15-11909-2015 (2015).

(2) Wenbo Sun, Gorden Videen, and Michael I. Mishchenko, "Detecting super-thin clouds with polarized sunlight," Geophy. Res. Lett. 41, 688-693, doi: 10.1002/2013GL058840 (2014).

---

## Author Comment (AC1) · 16 Feb 2016

Dear, Anonymous Referee #1,

The authors would like to appreciate for your careful reviews and comments to our manuscript. We response to your comments and show corrected parts of the manuscript as listed below. The referee comments are shown with Italic and the sentences with red color are added in the manuscript.

*1. When talking about CO2 slicing method, the authors should carefully introduce not only its benefit, but also its limitations in applications. E.g., can it be accurate when cloud is very thin, such as optical depth smaller than ~0.3? Does background surface temperature and atmospheric temperature profile and water vapor profile result in any uncertainty to the cloud retrieval? How significantly?*

As you pointed, the variables we have to assume to apply the method are large error sources. Menzel et al. (1992) had quantitatively investigated the influences to retrieved CTH and ECA from this method. Therefore, the manuscript is modified as below.

P.8, L25: ~ CTHs compared with lidar observations. In addition, uncertainty of the assumption such as surface skin temperature, temperature profile, and cloud multilayer structure are major sources of this method. Menzel et al. (1992) had quantitatively reported the influences for CTH retrievals from these assumptions.

*2. For Eq (1), in remote sensing practice, how can clear-sky R be obtained? This may be the most significant drawback of this method. The authors should justify the treatment.*

Clear sky radiance is calculated value from atmospheric transmittance, surface skin temperature, and surface emissivity. Transmittance is calculated using radiative transfer model, LBLRTM. Surface skin temperature and emissivity are from GSM-GPV and ASTER database noted in Sect. 2. The manuscript is modified as below.

P7. L21: ~ the spectral channel wavelength. Applying to satellite data, $t$ is calculated using LBLRTM at each layer level and $R^{clr}$ is calculated with the theoretical radiative transfer calculation from $t$, surface skin temperature, and surface emissivity based on GSM-GPV and ASTER database.

*3. In Section 4.1.2. "CTH was underestimated by slicing because of very thin cirrus near ..." why?*

The slicing method tends to estimate CTH as lower for optically thin clouds such as subvisible cirrus frequently occur in the Tropics and this fact was pointed in the previous researches. The pointed part is corrected as below.

P15, L12:  It is reported by previous researches that CTH of high clouds near tropopause are generally underestimate by the slicing method (e.g., Wylie and Wang, 1999; Wylie et al., 2007). This is mainly because the detection error of the slicing is relatively larger for optically very thin clouds such as subvisible cirrus as mentioned in Sect. 4.1.1.

*4. CALIPSO has significant errors for detecting optically thin stuff, especially during daytime, this should be discussed in the paper.*

Some sentences are added as below.

P6, L15: ~ 532 and 1032 nm. The background noises of CALIPSO observations are larger during daytime than those during night time because of sunlight contaminations. However, CALIPSO is able to detect optically thin cirrus of which optical thickness of 0.01 or less even during daytime (McGill et al., 2007). Therefore, it is considered that CALIPSO data are appropriate as validation data for this study.

*5. When discussing about historical background information about very thin cloud detection in Introduction, the data are not complete and updated. For references, please also cite the following 2 papers for latest developments in this important field.*
*(1) Wenbo Sun, Rosemary R. Baize, Gorden Videen, Yongxiang Hu, and Qiang Fu, "A method to retrieve super-thin cloud optical depth over ocean background with polarized sunlight", Atmos. Chem. Phys., 15, 11909-11918, doi: 10.5194/acp-15-11909-2015 (2015).*
*(2) Wenbo Sun, Gorden Videen, and Michael I. Mishchenko, "Detecting superthin clouds with polarized sunlight," Geophy. Res. Lett. 41, 688-693, doi: 10.1002/2013GL058840*

*(2014).*

The historical background about the optically thin cloud detection from space itself is not contained in the introduction. Therefore, the suggested two papers are referred in Sect. 2 as below.

P6. L3:  FTS and CAI are passive sensors using thermal or solar radiation. Although the recent studies report that optically very thin clouds are detectable with the data from passive sensors (e.g., Sun et al., 2014; Sun et al., 2015), the most accurate measurement of clouds and aerosols are using an active sensor, light detection and ranging (lidar), which emits a visible or near-infrared laser beam and receives their back-scattered components.

If you have any other additional comments, we are glad that you post them into the discussion page.

Sincerely,

Yu Someya
Atmosphere and Ocean Research Institute, the University of Tokyo
E-mail: y_someya@aori.u-tokyo.ac.jp

---

## Referee Comment (RC3) · Anonymous Referee #2 · 16 Mar 2016

General comments:

This paper presented an improvement to the CO2 slicing method for cloud detection using the hyper-spectral thermal infrared observations. By way of channel reconstruction, the measurement errors are reduced. Comparisons with the CALIPSO observations show that this improved algorithm apparently provides more accurate retrievals of the cloud top heights. This paper is well-constructed and concisely written. I recommend publication of this manuscript with the Atmospheric Measurement Techniques after minor revision.

Major issues:

[Figure]

* The authors used a lot of acronyms in this manuscript. Although using acronyms helps simply the long terms, too many of them also confuse people (and the authors themselves). I recommend the authors remove those acronyms that were used for only once, such as CLOUDIA (Page 3, line 10); LST (Page 14, line 7). Some acronyms are not defined before use, such as NIES (Page 5, line 21). As a result, it is highly recommended that the authors do a thorough check.

* I highly suggest the authors add a flow chart to help readers understand the procedures and steps of the retrieval process. Although the authors have described how the channels are reconstructed and optimized and how the method is applied to GOSAT, it is deemed necessary that a flow chart be added for better illustration.

* Page 12, line 24: "All the data for GOSAT and CALIPSO observed in January and July were analyzed." – The question is what is the temporal range of the data used? 2007 to present time? How many collocated observations are found? Please specify!

* About the differences in the retrieved cloud top height and cloud amount between this study and CALIPSO, how much of the bias is related to the differences in cloud optical properties used in the radiative transfer model? Also, the authors should specify what are the cloud optical properties they used in the radiative transfer calculation for retrieval purposes.

Minor problems:

* Page 2, line 23: "increasing 2.07 ppm" should be "increasing at 2.07 ppm"

* Page 3, line 14: "it enables to " should be "it is able to"

* Page 4, line 9: "calling" should be "called"

* Page 5, line 24-25: ". . . until July . . . from August" should be ". . . from January to July . . . from August to December"

* Page 6, line 28: "referred" should be "inferred"

\* Page 12, line 8: "the seen" should be "the scene"

\* Page 12, line 12: "compare the results" should be "compare with the results"

\* Page 13, line 23: "determine" should be "distinguish"

\* Page 14, line 12: remove "then"

\* Page 16, line 7: "generally agreement" should be "general agreement"

\* Page 17, line 24: "occur" should be "incur"

\* Page 18, line 10: "simultaneous studies" should be "simultaneous results"

---

## Author Comment (AC2) · 18 Apr 2016

Dear, Anonymous referee #2,

The authors would like to appreciate your careful reviews and detailed comments to our manuscript. We response to your comments and show corrected parts of the manuscript as listed below. Referee's comments were shown with *Italic* and the sentences with red color are added in the manuscript.

1.

*The authors used a lot of acronyms in this manuscript. Although using acronyms helps simply the long terms, too many of them also confuse people (and the authors themselves). I recommend the authors remove those acronyms that were used for only once, such as CLOUDIA (Page 3, line 10); LST (Page 14, line 7). Some acronyms are not defined before use, such as NIES (Page 5, line 21). As a result, it is highly recommended that the authors do a thorough check.*

We accepted your suggestion that the acronyms of CLOUDIA and LST should be removed. Although the acronyms such as VAS, MODIS, S-HIS, ASTER, IGBP, AER, HITRAN, and AIRS were used only once in the body of the manuscript, they had not removed because they are widely known in this study field. "NIES" was corrected as to "the National Institute of Environmental Studies (NIES)".

2.

*I highly suggest the authors add a flow chart to help readers understand the procedures and steps of the retrieval process. Although the authors have described how the channels are reconstructed and optimized and how the method is applied to GOSAT, it is deemed necessary that a flow chart be added for better illustration.*

We had added the flowchart of the simulation and retrieval processes as shown below;

[Figure]

Figure 1. Flow chart showing simulation-based channel optimizations and observed data analysis. White boxes are processes. The gray parallelograms are variables.

3.

*Page 12, line 24: "All the data for GOSAT and CALIPSO observed in January and July were analyzed." – The question is what is the temporal range of the data used? 2007 to present time? How many collocated observations are found? Please specify!*

The data used in this study were observed in 2010. The expression "in January and July" at Page 12, line 24 was corrected as to "in January and July in 2010". The number of collocated data was 123 for GOSAT and 316 for CALIPSO. Actually, this number is already specified on Page 17, line 4.

4.

*About the differences in the retrieved cloud top height and cloud amount between this study and CALIPSO, how much of the bias is related to the differences in cloud optical properties used in the radiative transfer model? Also, the authors should specify what are the cloud optical properties they used in the radiative transfer calculation for retrieval purposes.*

The $CO_2$ slicing method doesn't assume cloud properties in the radiative transfer calculations of the retrieval processes. They are assumed only in the simulations for the channel selection. However, the slicing method has to assume that the cloud emissivity is constant for all channels and clouds are infinitesimally thin. These assumptions can generate the CTH biases compared with CALIPSO observations. As mentioned at the end of Section 3.1 (Page.8, line 19~), Menzel et al. (1992) had reported that the errors associated with the assumption about the emissivity is negligible, but that about the cloud thickness can cause some biases especially for optically thin clouds. The previous studies (e.g., Hawkinson et al., 2005) reported the systematic biases of CTHs from the slicing method compared with the active sensor observations. Holz et al. (2006) reported that the slicing method detects the height where the integrated cloud optical thickness from the cloud top observed by the lidar is approximately 1.0. However, in the right panel of Fig. 11, the most clouds whose optical thickness less than 1.0 are precisely detected and the apparent negative biases of CTHs are not shown in this figure. Therefore, the biases related to the assumption about the cloud thickness could be not so large in the improved slicing method. The sentences were added into the text as below.

P. 17, line 20: ～detectable CTH. Holz et al. (2006) also reported that the slicing method detects the height at which the integrated cloud optical thickness from the cloud top observed by the lidar is approximately 1.0. However, most clouds for which the optical thickness from CALIPSO is less than 1.0 are detected precisely using the slicing method, as shown in this figure. This result demonstrates that the error associated with the assumption about the infinitesimally thin cloud is reduced by the reconstruction and optimization of spectral channels. In some cases～

The information about cloud particles assumed in the simulation studies was added in the sect. 3.3 as below.

P.10, line 14: ～and 6-15km. Water particles with modal radius of 8.0 μm were assumed for low clouds. Ice particles with modal radius of 20.0 μm were assumed for middle and high clouds. The cloud optical ～

5.
*Page 2, line 23: "increasing 2.07 ppm" should be "increasing at 2.07 ppm"*
*Page 3, line 14: "it enables to " should be "it is able to"*
*Page 4, line 9: "calling" should be "called"*
*Page 5, line 24-25: ". . . until July . . . from August" should be ". . . from January to July . . . from August to December"*
*Page 6, line 28: "referred" should be "inferred"*
*Page 12, line 8: "the seen" should be "the scene"*
*Page 12, line 12: "compare the results" should be "compare with the results"*
*Page 13, line 23: "determine" should be "distinguish"*
*Page 14, line 12: remove "then"*
*Page 16, line 7: "generally agreement" should be "general agreement"*
*Page 17, line 24: "occur" should be "incur"*
*Page 18, line 10: "simultaneous studies" should be "simultaneous results"*

All of the minor comments above were accepted to the manuscript.

Sincerely,

Yu Someya

Atmosphere and Ocean Research Institute, the University of Tokyo

E-mail: y_someya@aori.u-tokyo.ac.jp